# Social media usage reveals recovery of small businesses after natural hazard events

Robert Eyre [1], Flavia De Luca [2✉] & Filippo Simini [1,3✉]

The challenge of nowcasting the effect of natural hazard events (e.g., earthquakes, floods, hurricanes) on assets, people and society is of primary importance for assessing the ability of such systems to recover from extreme events. Traditional recovery estimates, such as surveys and interviews, are usually costly, time consuming and do not scale. Here we present a methodology to indirectly estimate the post-emergency recovery status (downtime) of small businesses in urban areas looking at their online posting activity on social media. Analysing the time series of posts before and after an event, we quantify the downtime of small businesses for three natural hazard events occurred in Nepal, Puerto Rico and Mexico. A convenient and reliable method for nowcasting the post-emergency recovery status of economic activities could help local governments and decision makers to better target their interventions and distribute the available resources more effectively.

[1] University of Bristol, Department of Engineering Mathematics, Bristol BS8 1UB, UK. [2] University of Bristol, Department of Civil Engineering, Bristol BS8 1TR, UK. [3] The Alan Turing Institute, London, UK. ✉email: flavia.deluca@bristol.ac.uk; f.simini@bristol.ac.uk

Post-event management is of interest to organisations that deliver aid, provide insurance, and operate in areas affected by natural hazards. The steps in disaster management are categorised into phases, with previous literature identifying anywhere from four to eight phases during the occurrence of a natural disaster[1]. Of the four phases used by the Federal Emergency Management Agency – mitigation, preparedness, response and recovery[2]—recovery is the least understood and investigated[3].

Multiple indicators for the recovery process of an area have been proposed[4], with many of them focusing on economic indicators of business activity thought to capture the long-term efforts of a region to return to a normal state. In particular, we refer herein to the recovery time as "downtime" relying on the definition within the Performance-Based Engineering framework; i.e., "the time necessary to plan, finance, and complete repairs on facilities damaged in earthquakes or other disasters"[5] and applying the downtime concept to small businesses.

The definition of actual business downtime, i.e., the time in which businesses are either closed or not running as expected, has been debated strongly in the literature. Chang's framework for urban disaster recovery[6] highlights three such definitions of recovery; returning to the same level of activity before the disaster, reaching the level of activity that would have been attained without the disaster happening, or attaining some stable level of activity that is different from these definitions.

Studies on long-term business recovery have been made using large-scale economic indicators, such as the number of reported businesses in Kobe City over a ten year period following the Kobe City earthquake in 1995[6]. These large-scale economic indicators are not as readily available or relevant for natural hazard events of a more moderate scale, inspiring the use of alternative statistics as predictors for business recovery, such as changes in pedestrian foot traffic[7], manually reporting on when businesses closed and opened[8] and changes in parking transactions[9], allowing for a much smaller scale to be studied. However, surveys and traditional direct data sources of businesses downtime have either been too costly or too inconvenient for widespread use.

Remote sensing has been shown to be vital in rapid damage detection and relief efforts after natural disasters. For example, the European Union's Copernicus Emergency Management Service and the Jet Propulsion Laboratory's Advanced Rapid Imaging and Analysis project use satellite imagery to provide assessment of regions devastated by different types of events. The main application of post-disaster satellite imagery is on emergency response and large-scale damage estimation[10], but as the frequency of image collection is increased, studies on long term recovery can be made. One such example is the study by Shermeyer on the long term recovery of the electrical infrastructure of Puerto Rico following Hurricane Maria[11].

Other efforts to estimate downtime using satellite imagery include counting the number of vehicles in imagery collected by Google Earth, correlating the presence of vehicles in business zones to the activity of businesses[12]. This method requires to have multiple satellite images of the region over time and may not be reliable in regions with developed public transport links, where the presence (or lack) of vehicles may not always correlate to small businesses' level of economic activity. A desirable secondary source of data for downtime estimation should be more readily accessible (easy and cheap to collect), have a high spatio-temporal resolution and, in the context of natural disaster, be rapidly obtainable. It is for these reasons that studies have turned to social media as a source of indicators for damage estimation after a natural disaster.

Social media data has been shown to be effective at rapidly detecting the presence of disrupting events such as earthquakes and floods, however the underlying process of social media use during these situations is not completely understood[13]. The micro-blogging service Twitter is often a source of related data, due to the nature of the platform—only allowing short 280 character maximum (often geo-tagged) posts promotes the exchange of informative data. Other social media and content sharing websites have also been shown to exhibit correlating behaviour to natural hazard events, such as the photo sharing website Flickr[14,15]. Many current social-media methods rely on sentiment analysis to filter messages by relevancy and quantify the severity of response to an event[13,16–20]. These methods offer rapid assessment of an event's infancy and are useful tools for understanding human behaviour during emergencies and to improve the delivery of hazard information in a region.

Recently private companies such as Facebook have started to look at how their own data can be used to support non-governmental and humanitarian organisations in understanding factors such as how people are displaced during natural disasters over longer time periods[21].

Mobile phone data has also been used to obtain insights on human behaviour during emergencies. Bagrow et al.[22] show that mobile phone data can be used to identify how human behaviour changes during periods of unfamiliar conditions, in a variety of situations ranging from bombings to festivals. In the context of natural disasters, the use of mobile phone records[23,24] have been used to measure population displacement after the 2015 Gorhka earthquake in Nepal, where negative inflows were recorded in Kathmandu for 56 days after the event[25]. One of the main limitations of mobile phone data is that it is usually not publicly available because of privacy concerns, hence obtaining mobile phone records for a specific region is not always possible.

In this paper we show that downtime can be estimated in real time using the public posts of local businesses on the social media site Facebook collected before, during and after a natural hazard event, without the need for natural language processing or semantic analysis. In particular, we consider three events of different types (two earthquakes and one hurricane) which occurred in countries with different indicators of socioeconomic development (Nepal, Mexico and Puerto Rico). The locations of the businesses collected in the three regions considered are shown in Fig. 1a–c and the respective time series of the total number of posts retrieved are shown in Fig. 1d–f.

## Results

**Facebook usage as an indicator of business downtime.** On the social media website Facebook, businesses can set up their own pages to advertise and interact with users, with approximately 65 million local businesses pages created as of 2017[26]. These business pages include the latitude and longitude of the business, allowing for the possibility of spatial analysis on the posts created in a given region. We apply our framework to estimate the average length of downtime of businesses in three different regions after three natural hazard events: the 2015 Gorkha Earthquake in Kathmandu, Nepal; Hurricane Maria in San Juan, Puerto Rico; Chiapas Earthquake in Juchitán de Zaragoza, Mexico (Fig. 1a–c). A rectangular bounding box containing each region of interest is specified. Publicly available business pages within the specified region were retrieved from Facebook via the Facebook Graph API, searching over the bounding box with different radii to maximise the number of found businesses. Once businesses' pages had been collected from Facebook, an additional set of queries were ran using the API to collect the time-stamps in which each business made a post. It should be noted that a visible post on a business page can be made by the owner of the business, or a member of the public. For the purpose of this study, we only

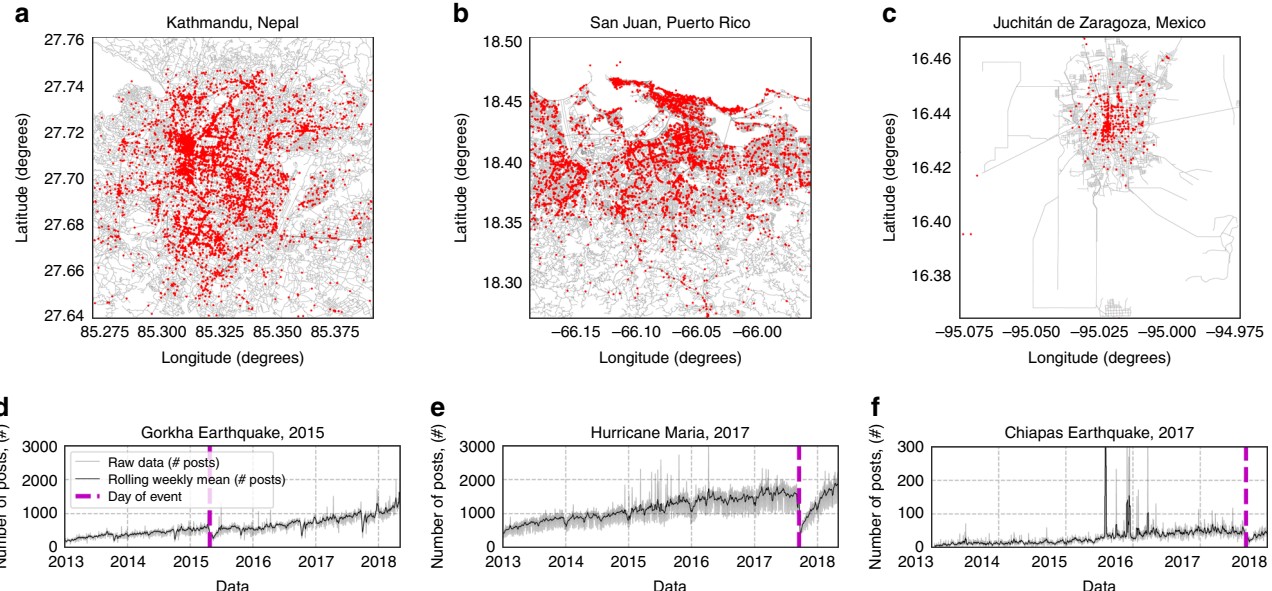

**Fig. 1 Businesses locations on Facebook with aggregate posting activity. a–c** Regions in which Facebook posts have been collected, each business highlighted in red, road networks collected from OpenStreetMap[39] (OpenStreetMap contributors, OpenStreetMap data is available under the Open Database License and licensed as CC BY-SA https://creativecommons.org/licenses/by-sa/2.0/) using OSMnx[40]. **d–f** Time series for the total number of posts made by all businesses in each region. The magenta vertical dashed lines denote the date of the natural hazard events.

collected the messages made by the owner of the page. The raw data consists of a list of posts and for each post two pieces of information are known: the date when the post was published and a unique ID of the business that made the post. The raw time series of the total number of posts in each region are shown in Fig. 1d–f. The number of businesses found in these regions and the number of Facebook posts collected are reported in Table 1.

To gauge the impact of a natural hazard event on the businesses within a specific region, we consider the businesses' posting activity on Facebook. Specifically, we compare the time series of number of posts of businesses in the time period after the event with the typical posting activity before the event.

Once the data has been collected, the framework is composed of two stages—data processing and downtime detection (Fig. 2). In the data processing stage, the time series of the posting activities are processed in order to remove trends and heterogeneous posting rates that are present in the raw data. In the downtime detection stage we define an automatic method to determine whether the posting activity presents anomalies, such as a posting rate significantly lower than the average rate, and use this to estimate the length of the downtime.

**Data processing**. We define the aggregated time series describing the behaviour of the entire region, $r(t)$, as the time series of the total number of posts made by all the businesses, $B$:

$$r(t) = \sum_{i \in B} x_i(t) \qquad (1)$$

where $x_i(t)$ is the number of posts made by business $i$ on day $t$. Typically, the raw time series $r(t)$ displays a reduced activity during a period of time following a natural hazard event (see for example Fig. 1d–f). However, it is difficult to estimate the significance and the length of the downtime from the raw time series because it is non-stationary. In particular, there is a general increasing trend in the average number of posts, due to widespread adoption to Facebook by local businesses over time.

Additionally, businesses have very different posting rates, with some being up to two orders of magnitude more active than the

average. Such very active businesses can introduce significant biases and lead to over or under estimate the activity level of the entire region, when the overall number of businesses is small. This is the case of the activity spikes in Fig. 1f in 2015, which are caused by a very small number of businesses.

To account for these issues, we develop a method to transform the raw time series into a detrended and rescaled series that allows us to clearly identify the periods of anomalous activity and measure their length. The methodology to process the time series is composed of four steps: the single business probability integral transform (PIT) and aggregation step, a shifting and rescaling step, a step in which we correct for the mean and variance, followed by the aggregated probability integral transformation.

**The single business probability integral transform**. The raw time series of the posts of each business is transformed into the time series of the corresponding "mid-quantiles". Formally this is obtained using a procedure similar to the PIT[27]. Let $x_i(t)$ be the number of posts of business $i$ on a given day $t$ and let $P_{X_i}(x)$ be the empirical cumulative distribution function (CDF) denoting the fraction of days of the current year when business $i$ made less than $x$ posts. We define the corresponding mid-quantile for $x_i(t)$ as $q_i(t) = [P_{X_i}(x_i(t)) + P_{X_i}(x_i(t) + 1)]/2$. We use the CDF of the current year, instead of considering the business' entire lifetime, to take into account long-term changes in a business' posting behaviour. Using the mid-quantile variable $q$ instead of the raw number of posts $x$ helps to reduce the bias due to outliers, for example days with an unusually high number of posts, and the bias caused by businesses with mean posting rates much higher than the average. Aggregating the transformed data, we have the time series

$$r_{PIT}(t) = \sum_{i \in B} q_i(t) \qquad (2)$$

**Shifting and rescaling**. After the PIT, the mid-quantiles $q_i(t)$ are expected to be uniformly distributed between 0 and 1. Under normal conditions (i.e. non-emergency and non-holiday periods),

**Table 1 Business data collected from Facebook in the three regions considered. The locations of the businesses are shown in Fig. 1a–c.**

| Region | Number of businesses | Number of posts |
|---|---|---|
| Kathmandu, Nepal | 11,818 | 1,182,878 |
| San Juan, Puerto Rico | 10,894 | 2,258,872 |
| Juchitán de Zaragoza, Mexico | 1728 | 62,999 |

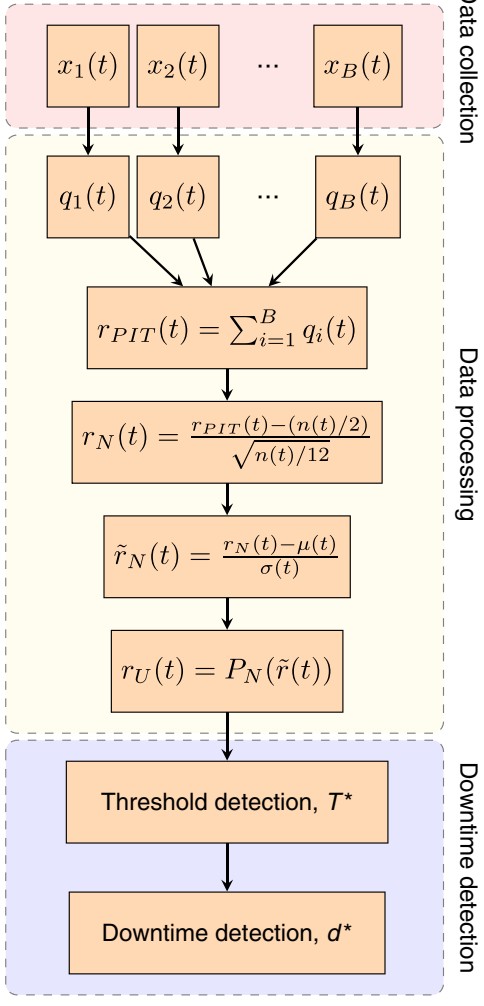

**Fig. 2 Schematic for the proposed framework to detect downtime of small businesses after natural hazard events.** Data collection: The time series of posting activity $x_i(t)$ for each business $i$ is collected. Data processing: The 'mid-quantiles' of each series $x_i(t)$ are computed to determine transformed individual time series $q_i(t)$ for each business $i$. The aggregate time series $r_{PIT}(t)$ is transformed by a shifting and rescaling to have mean zero and variance one ($\tilde{r}_N(t)$). The probability integral transform is then applied to form a final transformed time series $r_U(t)$ for the level of activity in the region. Downtime detection: Threshold $T^*$ is found using the elbow method to identify anomalous events. For a given event, the downtime length, $d^*$ is determined.

we expect businesses to have independent posting activities. Specifically, this is our definition of normal conditions: when businesses' posting activities can be considered independent random variables. Hence, the aggregated transformed data $r_{PIT}(t)$ of Equation (2) is expected to follow the Irwin-Hall distribution,

which is the distribution of the sum of $n(t)$ uncorrelated Uniform random variables between 0 and 1. Here $n(t)$ is the total number of businesses with a Facebook account on day $t$. When $n(t)$ is large, the Central Limit Theorem ensures that the distribution of $r_{PIT}(t)$ is well approximated by a Normal distribution with mean $n(t)/2$ and variance $n(t)/12$. Hence, by appropriately shifting and rescaling each day of $r_{PIT}(t)$ we define the time series

$$r_N(t) = (r_{PIT}(t) - n(t)/2)/\sqrt{n(t)/12} \qquad (3)$$

whose distribution is a Standard Normal with mean 0 and variance 1 for all days $t$. To perform the transformation of Equation (3) we have to estimate $n(t)$. The number of businesses with an active Facebook account on a given day is estimated by recording the first and last post date for each business $i$, denoted as $t_i^{(f)}$ and $t_i^{(l)}$ respectively. We make the assumption that a business remains active between these dates, plus a period of three more days to be sure that it has closed. This period was chosen to give an unbiased estimate of the number of active businesses at the tail end of the data collected. Formally, we estimate the number of businesses with an active Facebook account on day $t$ as $n(t) = |\{i : t_i^{(f)} \le t \le t_i^{(l)} + 3\}|$.

**Mean and variance correction**. The assumption that the businesses' posting activities are independent and the estimate of the number of businesses on Facebook $n(t)$ are reasonable, but may not be perfectly exact. In fact, a small correlation between businesses' posting activity may be present even during normal (non-emergency and non-holiday) conditions and this would affect the variance of $r_{PIT}$. On the other hand, deviations in the estimate of $n(t)$ could also change the mean of $r_{PIT}$. To correct for these possible sources of bias, we fit and remove a linear trend from $r_N(t)$, so that its mean becomes zero, and we divide it by its standard deviation, so that its variance becomes one. The resulting corrected time series, $\tilde{r}_N(t)$, for Kathmandu is shown in Fig. 3c and those for the other regions considered are shown in Supplementary Figs. 1 and 2.

**Aggregated probability integral transform**. Finally, we apply the Probability Integral Transform to the aggregated time series of normally distributed variables, $\tilde{r}_N(t)$, to obtain a time series of variables uniformly distributed between 0 and 1. This is done using the CDF of the Standard Normal distribution, $P_N$:

$$r_U(t) = P_N(\tilde{r}_N(t)) \qquad (4)$$

The final time series $r_U(t)$ for Kathmandu is shown in Fig. 3d and those for all the regions are shown in Fig. 4. The four steps of the data processing for all the regions considered are shown in the Supplementary Figs. 1 and 2.

To summarise, the outcome is a transformed time series without any long-term trend and without bias towards highly active businesses. Specifically, the methodology removes the long-term nonlinear trend of aggregated posting activity, while retaining the dynamics at short and medium time scales (i.e., weeks, months), and equally weights the contribution of each business avoiding to over-represent the activity of businesses with higher posting rates.

**Downtime detection**. The length of downtime of a system is generally defined as the length of time during which it is not operating as expected, i.e., the level of a given indicator of the system's performance or activity is significantly reduced[6]. In the context of estimating the downtime of small businesses in a region, we define the indicator of aggregated activity for the region as the transformed time series of the number of posts made by all businesses, $r_U(t)$. The transformed time series $r_U(t)$ is a good

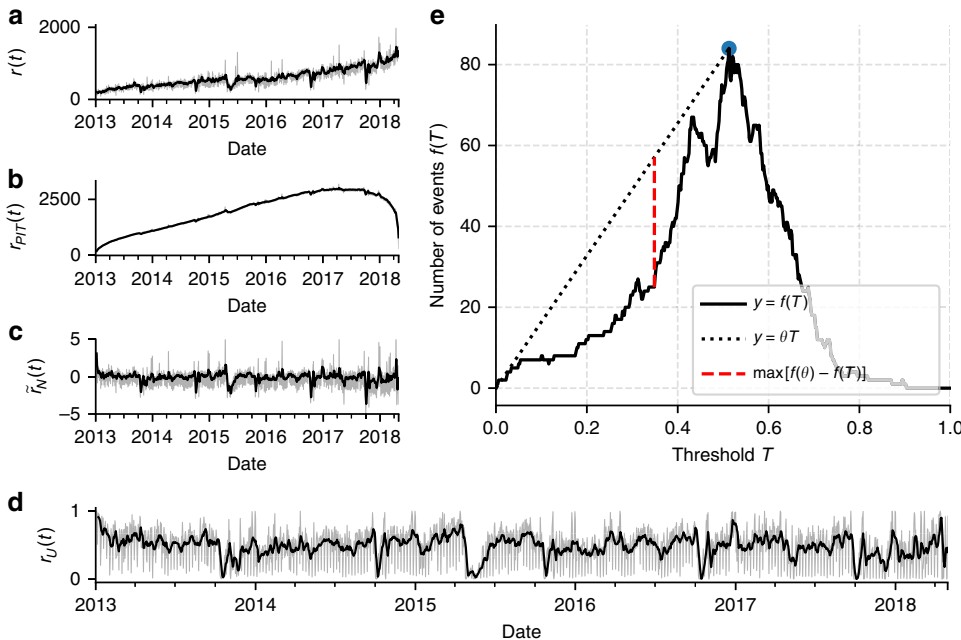

**Fig. 3 Steps of the methodology to estimate the downtime of small businesses applied to Kathmandu, Nepal. a** Time series of the total number of posts of all businesses ($r(t)$), shown with weekly rolling mean (black solid line). **b** Single business Probability Integral Transformed data ($r_{PIT}$). **c** Data is shifted and rescaled according the number of active businesses. **d** PIT applied on the aggregated and transformed time series. **e** Downtime detection using the elbow method.

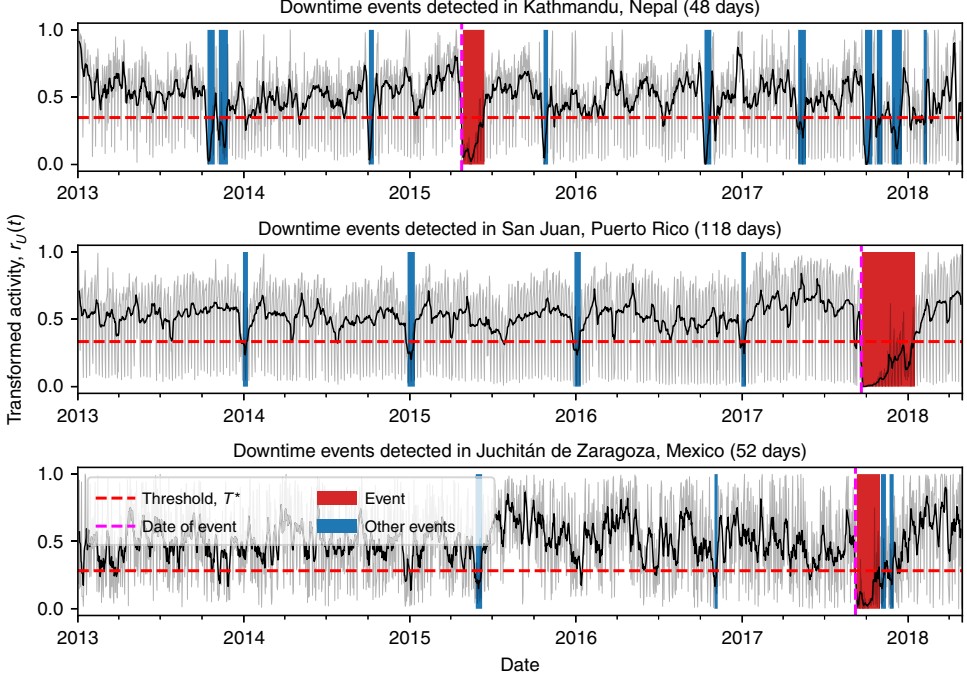

**Fig. 4 Detected downtime events.** Downtime events longer than seven days detected in **a** Kathmandu, Nepal, **b** San Juan, Puerto Rico, **c** Juchitán de Zaragoza, Mexico. Transformed posting activity data, $r_U(t)$ (Eq. (4)), is shown in grey, its one week rolling mean is shown in black and the thresholds $T^*$ estimated with the elbow method are marked with dashed horizontal red lines. The detected periods of downtime are highlighted with vertical red bars (the natural hazard related event) and blue bars (other events). The date of the event in each region is marked by a dashed vertical magenta line.

indicator of the aggregated activity of all businesses because it has no long-time (yearly) trend and it equally weights the activity of each business. To identify periods of significantly reduced aggregated activity (i.e., downtime) we propose the elbow method, which allows to determine a threshold on the level of activity, $T^*$, where activity fluctuations become unusually large. The length of

downtime is then estimated as the time taken for $r_U(t)$ to return to a normal level of activity, above the threshold $T^*$.

In normal conditions the transformed time series $r_U(t)$ is expected to fluctuate uniformly within 0 and 1. In anomalous conditions, for example when all businesses are closed after a natural hazard event, their posting activities are instead highly

correlated. In these cases, large fluctuations will be present in $r_U(t)$, which are the signature of an anomalous behaviour (i.e., a period of downtime) that we would like to detect using the elbow method described in the following. The length of downtime is estimated considering the rolling weekly mean of the transformed time series $r_U(t)$. This is primarily to account for differing posting behaviour over the course of a week (e.g., weekdays vs. weekends).

**The elbow method**. We define an event as a period of time when the weekly mean of $r_U(t)$ is below a given threshold value, $T$, for more than 7 consecutive days. The value of the threshold is set using a method aimed at detecting all periods when posting activity is significantly below the characteristic range of normal activity fluctuations. This value is found using the following elbow method. First, the number of potential downtime events are recorded at multiple threshold values; the number of events for a given threshold value is denoted as $f(T)$ and it is shown in Fig. 3e for Kathmandu. Second, we identify the threshold value, $T^*$, at which the number of events begins to rapidly rise. This point marks the transition between the size of activity fluctuations due to anomalous events and the characteristic size of normal activity fluctuations. We define $T^*$ as the abscissa of the point on the "elbow" of function $f(T)$, i.e. the value of $T$ that maximises the distance between $f(T)$ and the line connecting the origin to a point on $f$ such that its slope, $\theta$, is maximum (see Fig. 3e): $T^* = \arg\max_T(\theta T - f(T))$. The dashed vertical line in Fig. 3e denotes the value of $T^*$ for Kathmandu. The length of downtime is estimated as the number of days from the event date in which the rolling weekly mean of the transformed activity is below the threshold $T^*$. Given that we are using a weekly rolling mean to detect the end and duration of the downtime, we expect our estimates to have an uncertainty of around one week. Examples of downtime detected using this method are shown as red vertical bars in Fig. 4, where the thresholds $T^*$ are marked with red dashed horizontal lines.

After applying the whole framework to the three regions that we have considered, we see the events shown in Fig. 4; the start date, end date and the length of downtime of all the events detected are reported in Supplementary Tables 1–3. We find downtime in Kathmandu (48 days), Juchitán de Zaragoza (52 days) and San Juan (118 days) on the dates of their respective natural hazard events. The following sections describe the three natural hazard events and discuss the validation of our downtime estimates. Finally, we show that our methodology can accurately estimate the recovery status of businesses in real-time during the weeks after natural hazard events.

**Observed downtime in Kathmandu**. The 2015 Gorkha earthquake was one of the worst natural hazard events in Nepalese history, killing around 9,000 people, and injuring nearly 22,000 people. It occurred on April 25th, with a magnitude of Mw 7.8. Its epicentre was located 77 km North West of Kathmandu (east of the Gorkha district). Hundreds of aftershocks were reported following the earthquake, five of which registered Mw above 6. The most notable aftershock occurred on the 12th of May, killing a further 200 people and injuring a further 2500. 8.1 million Nepalese citizens were thought to be affected by the earthquake[28]. The event affected the whole country, with notable damage to historical buildings in the northwestern part of Kathmandu[29].

Our method estimates a downtime of 48 days over the entire city of Kathmandu after the Gorkha earthquake in 2015 (see Fig. 4a). We detect shorter downtimes in other times of the year, further supporting the validity of our method (see Supplementary Table 1). Indeed, the majority of the Nepalese people (84.7%) practice Hinduism and celebrate multiple festivals throughout the

year. The main festival, Dashain, is celebrated for two weeks in September or October and is of such importance to the religious and cultural identity of the country, that businesses and other organisations are completely closed for 10 to 15 days to celebrate. As a result, we detect downtime during Dashain for each year in which we have data.

**Observed downtime in San Juan**. Puerto Rico, along with Florida and the US Virgin Islands, was hit by a Category 5 hurricane on September 20th, 2017 causing significant damage to infrastructure[30], affecting 100% of the population[31], with an estimated 45% of islanders being without power for three months after the event[32]. Hurricane Maria is estimated to have caused $94 billion in damage, with as many as 60,000 homes still lacking roofs as of December 2017[33]. As of August 2018, the Federal Emergency Management Agency (FEMA) have approved 50,650 loans to small businesses proving $1.7 billion to aid in recovery, in total obliging $21.8 billion in relief efforts[31].

Our method estimates a downtime of 118 days in San Juan after hurricane Maria (see Fig. 4b). This can be split into downtime from the hurricane (104 days), and downtime from the Christmas to New Years period (14 days). In fact, in San Juan, we find other shorter periodic downtime periods each year between Christmas and New Year (see Supplementary Table 2). Compared to Nepal, Puerto Rico has a majority Christian population, which explains the downtime observed during the Christmas period.

**Observed downtime in Juchitán de Zaragoza**. The 2017 Mw 8.2 Chiapas earthquake was the second strongest earthquake recorded in Mexico's history (and most intense globally in 2017), triggered in the Gulf of Tehuantepec, just off the south coast of Mexico. Multiple buildings in the city closest to the epicentre, Juchitán de Zaragoza, were reduced to rubble with people sleeping outdoors due to fears of further damage[34]. The main hospital in Juchitán also collapsed causing severe disruption to medical services in the area, with 90 reported dead in Mexico. Food shops were also affected, with prices rises due to closures, and fears from looting causing more closures. According to local authorities, roughly 30% of houses were damaged in the earthquake. This is probably aggregated by the lack of earthquake resilient houses, predominantly made of adobe block with concrete bond beam and mud wall constructions[35].

Our method estimates a downtime of 52 days in Juchitán de Zaragoza after the Chiapas earthquake (see Fig. 4c). Other shorter downtime periods are detected that may be due to religious and national holidays, but these attributions are uncertain due to the noisier time series (see Supplementary Table 3).

**Validating observed downtimes**. We use a variety of validation sources of the actual downtime (see Table 2): text analysis, surveys, official reports and scientific publications. We find that they all agree on similar downtime estimates that are compatible with the estimates of the proposed framework. Each validation source is described in the Methods.

**Downtime detection in real-time**. The proposed system can be applied in real-time giving estimates of the recovery status during the weeks immediately after an event. We simulate the collection of data in real time by cropping our data in the weeks following the event and we calculate $d_{RT}(t)$, the real-time estimate of the downtime $t$ days after the event, using just the posts published until day $t$. Results for Kathmandu are shown in Fig. 5, for the other regions see Supplementary Figs. 3 and 4. To evaluate the accuracy of the real-time estimate, we measure the root mean squared distance (RMSD) between $d_{RT}(t)$ and the ideal

**Table 2 Validation sources for the proposed framework. Downtimes estimated with the proposed framework (Estimated downtime) for the three natural hazard events considered, along with the downtimes reported by the various validation sources described in Methods.**

| Region and Event | Source | Downtime Length |
|---|---|---|
| Kathmandu, Nepal | **Estimated downtime** | **48 days** |
| Gorkha Earthquake | Business surveys, from[12] | 41 days |
| | Kathmandu Post Disaster Needs Assessment[41a] | 37 days |
| | Mobile phone data, from[25] | 56 days |
| | Facebook posts text analysis ($n = 299$) | 51 days |
| San Juan, Puerto Rico | **Estimated downtime** | **118 days** |
| Hurricane Maria | Satellite imagery, from[11,37] | 134 days |
| | Puerto Rico Tourism Company[b] | 97 days |
| | U.S. Energy Information Administration[c] | 128 days |
| | Facebook posts text analysis ($n = 755$) | 71 days |
| Juchitan de Zaragoza, Mexico | **Estimated downtime** | **52 days** |
| Chiapas Earthquake | Facebook surveys ($n = 16$) | 63 days |
| | Facebook posts text analysis ($n = 19$) | 55 days |

[a]"The earthquakes and series of continuing aftershocks led to the complete closure of schools and colleges for 37 days" (https://www.nepalhousingreconstruction.org/sites/nuh/files/2017-03/PDNA%20Volume%20A%20Final.pdf).
[b]https://tourism.pr.gov.
[c]https://www.eia.gov/electricity/monthly/.

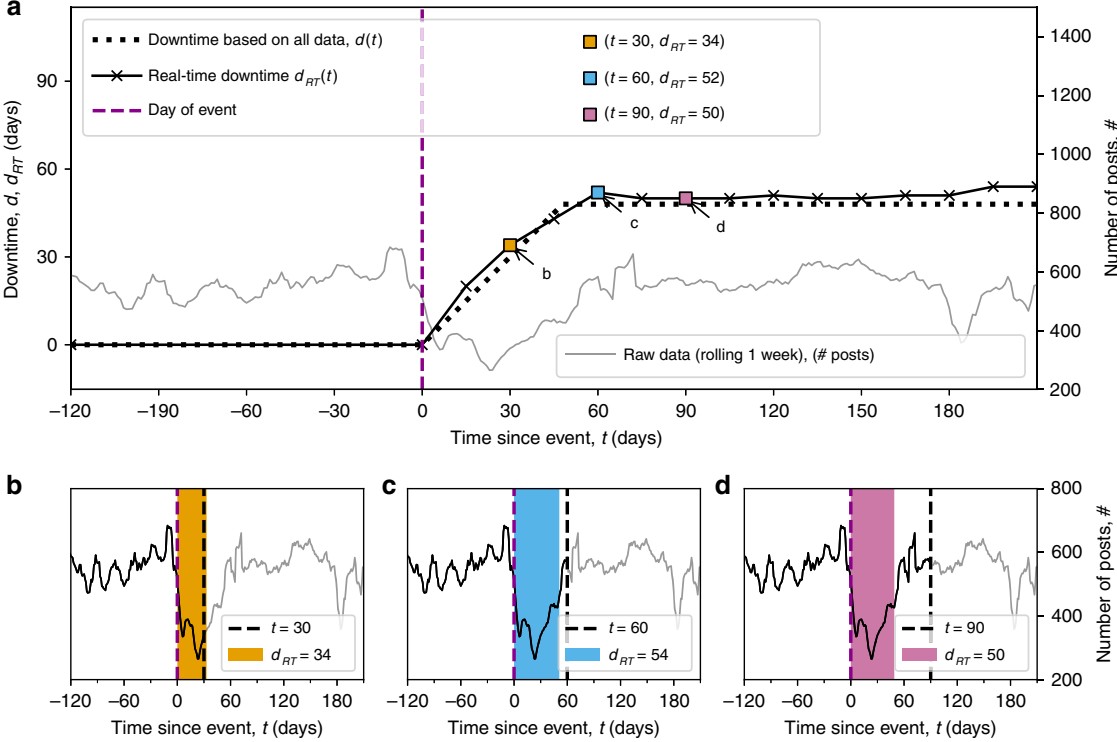

**Fig. 5 Downtime detection in real-time ($d_{RT}(t)$) for Kathmandu, Nepal. a** Data are cropped at regular intervals to simulate real-time data collection (crossed markers along solid line). Square markers at $(t, d_{RT}(t))$ indicate the real-time estimates for data cropped at $t = 30$, 60, 90 days. Dotted line indicates the ideal downtime, $d(t)$, estimated using all data. **b–d** Black solid lines denote the cropped data used to estimate downtime at $t = 30$, 60, 90 days after the event (dashed magenta line) respectively. The cutoff $t$ is shown with a black dashed line. The estimated downtimes $d_{RT}(t)$ are shown with coloured areas.

downtime, $d(t) = \min(t, d^*)$, where $d^*$ is the downtime estimated using all data ($d^* = 48$ days for Kathmandu). Computing the RMSD $= \sqrt{\sum_{t=0}^{d^*} (d_{RT}(t) - d(t))^2 / d^*}$ from the date of the event $t = 0$ until $t = d^*$, i.e. for a time period of the length of the actual downtime from the date of the event, we obtain 1.94, 1.34,

3.16 days for Kathmandu, San Juan and Juchitán de Zaragoza respectively. In all cases the errors are within the method's accuracy of plus/minus one week, demonstrating the possibility to obtain accurate real-time estimates of downtime. The error is larger in Juchitán de Zaragoza because of the larger fluctuations of the estimates due to a lower number of businesses.

## Discussion

A framework has been proposed to determine the recovery status of small businesses in the aftermath of natural hazard events through the analysis of the businesses' posting activity on their public pages on the social media site Facebook. The framework relies on the assumption that businesses tend to publish more posts when they are open and fewer when they are closed, hence analysing the aggregated posting activity of a group of businesses over time it is possible to infer when they are open or closed.

The methodology is tested in three case studies, where it successfully estimates the downtime of businesses in three urban areas affected by different natural hazard events: the 2015 Gorkha earthquake in Nepal, the 2017 Chiapas earthquake in Mexico and the 2017 hurricane Maria in Puerto Rico. We validate our estimations using various data sources: field surveys, official reports, Facebook surveys, Facebook posts text analysis and other studies available in literature.

Our results demonstrate that the posting activity of small businesses on social media can be used to estimate the recovery status of regions hit by natural hazard events in real time. The methodology has general applicability: it works for different types of natural hazards (earthquakes and hurricanes) and for regions in different continents, in both developed and developing countries. The proposed framework offers several advantages with respect to other methods to estimate long-term post-disaster economic recovery: it is cheaper and more scalable than traditional field surveys, it is privacy-compliant because it is not based on sensitive personal data, and it is significantly easier to implement than methods based on text and sentiment analysis of social media content.

The proposed methodology has the potential to be of interest to various stakeholders aiming to provide support to regions hit by disasters, including local and national governments, international financial institutions, and humanitarian organisations. A possible application is the automatic identification of areas whose recovery appears to be lagging behind, in order to deliver help and support where it is most needed.

The construction of an automatic system to detect downtime in various regions is possible because of the method's unique features, which make the proposed framework highly scalable: global coverage, as the method has no intrinsic geographic coverage limitation and is applicable to any region where enough businesses are active on social media; and easy implementation, as it generates real-time downtime estimates completely automatically, without the need to make any ad-hoc or manual adjustment to the algorithm.

Like in all studies based on social media data, it is important to remark that such data may not necessarily represent all of a population evenly[36]. In fact, some businesses may not have a social media account and the group that does have it may not be representative of the entire business population. When using the proposed methodology to draw conclusions on the degree of impact in different regions, particular attention should be taken to ensure to have a representative sample of the businesses in the region. Should that not be the case, the downtime for the type of businesses not represented in the data set should be estimated using other methods.

The proposed methodology for the detection of anomalous events in non-stationary time series is of general applicability to all cases where the main signal is composed by the aggregation of a large number of individual components, for example phone users' calling activities or internet users' visits to web pages.

## Methods

**Facebook posts text analysis**. Text analysis of social media content has been previously used to assess the impact of natural hazards on different communities[13,36]. We employ a text analysis method to obtain information about the reopening date of the businesses looking at the content of their messages on Facebook. We compute the average reopening times reported by firstly sampling posts following each natural hazard event. We look at posts published up to five months from the date of the event (six months in San Juan). We sampled 40,946 posts in Kathmandu, 94,611 posts in San Juan and 4536 posts in Juchitán de Zaragoza respectively. Next we determinined a set of keywords that indicate that a business has reopened. We select all messages containing the words: 'open again', 'reopen', 'normal' and 'regular', in English, Spanish and Nepalese. For each business in our sample, we estimate its reopening date as the date of the first post that contains one of the selected keywords. The overall average downtime is defined as the average of the durations it took for businesses to reopen.

This method is applicable to all three regions considered. Our downtime estimates using Facebook posts text analysis are: 51 days for Kathmandu, 71 days for Puerto Rico and 55 days for Juchitán de Zaragoza. The text analysis' results for Kathmandu and Juchitán de Zaragoza are very similar to the lengths of downtime estimated with our method, while a shorter downtime is obtained for San Juan. Some limitations of the text analysis for estimating the downtime are discussed in Supplementary Note 1.

**Research field data**. Surveys from four areas in Kathmandu were used to verify the downtime found in this region. These surveys were part of a research field mission in 2016 to examine the extent of the earthquake damage, with regions chosen that vary in population density, construction methods and traffic concentration[12]. These surveys record the length of downtime for each business, along with the reason for closure. The mean downtime reported by the businesses surveyed is 41 days (see Supplementary Table 4, Supplementary Fig. 5).

**Official Nepalese government reports**. Following the earthquake in Kathmandu, the National Planning Commission of Nepal released two "Post Disaster Needs Assessment" reports. The first report (Volume A: Key Findings) reports 37 days of closure for schools and colleges in the region.

**Mobile phone data**. A research study[25] used mobile phone trajectories to reconstruct mobility flows among cities in Nepal following the earthquake. This study reports that after approximately 56 days, the number of people returning to Kathmandu was greater than the number of people leaving Kathmandu.

**Satellite imagery data**. A research study[11] measured the change of brightness in satellite images over time to assess electrical and infrastructure recovery around Puerto Rico. We use data[11,37] provided by the author of this work to validate our methodology by applying the elbow method to this data, returning 134 days of downtime in which brightness was significantly reduced (see Supplementary Fig. 6a).

**Tourism data**. To validate the downtime found in San Juan, historical tourism data has been retrieved from the Puerto Rico Tourism Company[38], listing the cruise passenger movement in port in Old San Juan. Applying the elbow method to this data returns 97 days of downtime in which tourist arrivals were significantly reduced (see Supplementary Fig. 6b).

**Energy usage information**. The Energy Information Administration provides detailed usage statistics of energy in the United States. We look at the amount of bought electricity by the state of Puerto Rico and apply the elbow method to this data, returning 128 days of downtime in which energy usage was significantly reduced (see Supplementary Fig. 6c).

**Facebook surveys**. We sent a survey to 52 of the businesses found via Facebook in Juchitán de Zaragoza, asking the question "Were you affected by the earthquake, and if so, for how long were you closed?". With a response rate of 30% (16 responses), the average closure time reported is 63 days.

**Sensitivity analysis**. The reported downtime for each region was calculated using all of the collected businesses that had posted at least once. We tested the sensitivity of our method to the overall number of businesses considered, including only businesses that posted one year before the event (see Supplementary Note 2, Supplementary Fig. 7). We observe that the methodology gives consistent estimates when the number of businesses considered is large (i.e. thousands), whereas the length of downtime might be underestimated if the number of businesses considered is small. We verified this in Nepal and Puerto Rico, by randomly sampling a subset of businesses and computing the average length of downtime over 1000 realisations as a function of the sample size (see Supplementary Fig. 8). Results are more consistent for large samples because of the law of large numbers: as the number of businesses increases, the empirical averages of the variables that we define in our methodology (e.g., $r(t)$, $r_{PIT}(t)$) tend to their expected values and the

downtime's estimate becomes more robust. Moreover, because of the central limit theorem, the rescaled distribution $r_N(t)$ is well approximated by a normal distribution only when the number of businesses is large. However, establishing a precise relationship between the number of businesses and the average downtime is difficult because that estimate depends not only on the number of the businesses but also on other factors, such as their geographic density and mean posting rate. Indeed, to assess to what extent the estimate of downtime depends on the posting activity of the businesses, we filtered businesses by their daily posting rate and by the total number of posts they have made. The results reported in the Supplementary Tables 5–7 show that the overall downtime of the region is not affected by the filtering, except in cases of very high thresholds. In particular, we note that we can obtain an accurate estimate of the downtime with a small number of businesses (few hundreds) that post frequently (e.g., more than once per week).

## Data availability

Data to reproduce the results presented in this paper can be found at https://github.com/roberteyre/Business-Recovery. In order to preserve the privacy of the businesses considered, we applied the following procedure to anonymise the data. First, we substituted the real Facebook ID of each business with an integer number. Second, we added a random noise uniformly distributed in $[-2, +2]$ days to the actual date of each post. While this procedure prevents the simple reidentification of a business based on the exact dates of its posts, it does not change the posting pattern on temporal scales longer than one week, which is the time resolution of our study. As a result of the addition of random noise to the posting date, the results may be slightly different from those reported here.

## Code availability

Code to reproduce the results presented in this paper is available at https://github.com/roberteyre/Business-Recovery.

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

## Acknowledgements

The authors would like to thank Jacob Shermeyer for providing the light data used to validate the downtime estimation in San Juan, Raffaele De Risi, Luca Lombardi and Rama Mohan Pokhrel who were part of the field survey team in Kathmandu in November 2016 and Jennifer Kurton who helped processing the Kathmandu survey data. RE is grateful for the support received by EPSRC (EP/N509619/1) and the 2017/18 Cabot Institute Innovation Fund. FDL acknowledges the support of the Leverhulme Trust (RPG2017-006, GENESIS project) and EPSRC for funding the field data collection in Kathmandu (EP/P510920/1). FS is supported by EPSRC (EP/P012906/1).

## Author contributions

All authors contributed to the conception and design of the study. F.D.L. conducted the field survey, R.E. collected the social media data and analysed the results, F.S. and F.D.L. supervised the project. All authors wrote and reviewed the manuscript.

## Competing interests

The authors declare no competing interests.
