## [Peer Review File · Nature Communications]

Reviewers' comments:

Reviewer #1 (Remarks to the Author):

This paper tackles a very interesting and timely topic: estimating the post-emergency downtime of small businesses by examining their online posting activity through the social media website Facebook. The reviewer feels that the work is quite novel, however, a major review is needed for clarity in the methodology and application of the proposed framework. These points are further discussed below along with some additional comments meant to improve the manuscript.

Specific Comments:

- 1) The title is not descriptive of the work done. There should be some mention of estimating the downtime of small businesses. The current title is much broader and leaves the impression that the authors were looking into full scale societal recovery (of which business interruption is one part of.)
- 2) It is clear that the authors' methodology could be used as an indicator for small business' recovery following emergencies. The authors then go on to claim that the methodology can be useful for prediction. This is unclear to the reviewer since the current methodology seems to be applied to only historical data. More detail is needed if the methodology should be applied to prediction.
- 3) Presenting the methodology and data section before presenting the results would help with clarity.
- 4) Whenever the authors would like to refer to a section, the section number seems to be missing. For example, page 3, paragraph 2.
- 5) On page 5, line 1, do the authors mean that they detect downtime after festivals that follow Dashain?
- 6) In the sensitivity analysis section, the authors note that the methodology is more consistent when n is large. Isn't this a direct result of the authors using the central limit theorem for their analysis?
- 7) Figure 3 is not clear. The legend does not provide descriptions of the different markers nor is it clear in the left-hand figure where the three time series are and how the prediction is changing.
- 8) To apply the author's methodology, how long does a business have to be in operation prior to the event? This seems to be an important since a newer business might not have enough data to apply this methodology accurately (given that the time series of each business is independent as discussed in comment 8.)
- 9) In data processing on page 8, the authors note that there are seasonality effects. How, exactly, is the seasonality dealt with? The authors' correction seems to be only for a linear trend.
- 10) In step 2 on page 9, the authors are using the Irwin-Hall distribution (and eventually the central limit theorem) which requires uncorrelated random variables. How are the authors justifying the variable independence? The correction discussed in step 3 does not adjust for correlation, only a linear trend. Moreover, this method hinges on the idea that the businesses' Facebook activity is highly correlated after an emergency.

Reviewer #2 (Remarks to the Author):

This manuscript provides analysis of Facebook posting volumes to infer the time of closure of businesses after natural disasters. The focus of the work is primarily on an algorithm to detect changes in time series of the number of posts made by businesses in a specified geographic area. While the topic is interesting, the results feel anecdotal and the manuscript does not give substantive insight into general conclusions from this work. A few specific concerns are noted below:

-It is unclear what exactly is to be concluded from the study. The first paragraph concludes with some speculation that the results might be useful to "target their interventions" and "distribute the available resources more effectively", but the metrics considered here don't link in any obvious way to those topics. Rather, the focus is primarily on the proposed algorithm and its application to three time series.

-It is unclear how the specific regions of study were determined here. Disaster impacts vary spatially, and presumably there are hard-hit regions and less impacted regions in each study area, but it is unclear what degree of impact was seen in each of these study areas, or how the closure time would be affected by changes in the study extent. Palen and Anderson (2016) note the importance of this decision--a reference that the authors should review in general.

-It is unclear what it means for a region to have a single "downtime" number. The recovery process is a continuous one, and certainly the businesses are re-opening individually over time. Without a well defined metric for what "downtime" is, the interpretation of the reported results, and the opportunities for validation of the results, are quite difficult.

-Related to the prior comment, the benchmarking of these results is weak. While it is admittedly difficult to get ground truth of closure times, it is concerning that each case study has a different method for determining an actual downtime--presumably different methods applied to a single disaster event would produce differing downtime estimates.

The benchmarks are also based on extremely limited data (e.g., 16 survey responses), or very indirect measures (e.g., using tourist arrival numbers). It seems feasible to do something more direct and reproducible such as checking the posts for messages of "we are open again" or similar. The data sets are not huge, and sampling schemes or text analysis algorithms could help.

Palen, L., and Anderson, K. M. (2016). "Crisis informatics—New data for extraordinary times." *Science*, 353(6296), 224–225.

Referee #1

This paper tackles a very interesting and timely topic: estimating the post-emergency downtime of small businesses by examining their online posting activity through the social media website Facebook. The reviewer feels that the work is quite novel, however, a major review is needed for clarity in the methodology and application of the proposed framework. These points are further discussed below along with some additional comments meant to improve the manuscript.

Response: We thank the referee for acknowledging the interest and novelty of our work and for the relevant comments and suggestions that helped to greatly improve the manuscript.

Point 1. *The title is not descriptive of the work done. There should be some mention of estimating the downtime of small businesses. The current title is much broader and leaves the impression that the authors were looking into full scale societal recovery (of which business interruption is one part of.)*

Response: Following the referee's suggestion we changed the title into "Social media usage reveals how small businesses recover after natural disasters in urban areas". The new title is more descriptive of the analysis presented in the work, specifying that the proposed methodology focuses on estimating the recovery of small businesses in urban areas.

Point 2. *It is clear that the authors' methodology could be used as an indicator for small business' recovery following emergencies. The authors then go on to claim that the methodology can be useful for prediction. This is unclear to the reviewer since the current methodology seems to be applied to only historical data. More detail is needed if the methodology should be applied to prediction.*

Response: We thank the referee for giving us the opportunity to clarify this important point of our methodology in the revised manuscript. Our algorithm can be used to estimate economic recovery after emergencies in real time, as demonstrated in the paper's section "Downtime detection in real-time". Specifically, we show that our methodology can effectively estimate the length of downtime with an accuracy of one week at any time during and after the emergency. We demonstrate this in Figure 5, where we report the downtime computed on a given date after the event (e.g. 30 days since the event) where we only take into account posts published up to that date. Results show that the real-time estimate of the downtime is always close to the estimate done using all data, confirming that our methodology can reliably estimate downtime *during* emergencies.

To address the confusion due to the misuse of the word *prediction* in the original manuscript, in the revised version we have replaced the word *predict* with the word *estimate*.

Point 3. *Presenting the methodology and data section before presenting the results would help with clarity.*

Response: As suggested, we have swapped the two sections, placing the Methods section before the Results and Discussion sections. We agree with the referee that presenting the methods before the results greatly improves the clarity of the manuscript.

Point 4. *Whenever the authors would like to refer to a section, the section number seems to be missing. For example, page 3, paragraph 2.*

Response: This was a formatting error and has now been fixed.

Point 5. *On page 5, line 1, do the authors mean that they detect downtime after festivals that follow Dashain?*

Response: This line was meant to state that we detect other festivals, not just Dashain (the main festival in Nepal). We removed this line to avoid confusion. All downtimes we detect are listed in Tables 2-4 in the Supplementary Materials.

Point 6. *In the sensitivity analysis section, the authors note that the methodology is more consistent when n is large. Isn't this a direct result of the authors using the central limit theorem for their analysis?*

Response: We agree with the referee that according to the central limit theorem, the rescaled distribution $r_N(t)$ is well approximated by a normal distribution as the number of businesses increases. In general, we believe that the methodology is more consistent when we consider a large number of businesses because of the law of large numbers: as the number of businesses increases, the empirical averages of the variables that we define in our methodology (e.g., $r(t)$, $r_{PIT}(t)$, ...) tend to their expected values and the downtime's estimate becomes more robust.

To clarify this point, we added the following paragraph to Section 3.1 "Sensitivity analysis":

Results are more consistent for large samples because of the law of large numbers: as the number of businesses increases, the empirical averages of the variables that we define in our methodology (e.g., $r(t)$, $r_{PIT}(t)$) tend to their expected values and the downtime's estimate becomes more robust. Moreover, because of the central limit theorem, the rescaled distribution $r_N(t)$, computed in Step 2 of our methodology, is well approximated by a normal distribution only when the number of businesses is large.

Point 7. *Figure 3 is not clear. The legend does not provide descriptions of the different markers nor is it clear in the left-hand figure where the three time series are and how the prediction is changing.*

Response: The figure has been completely revamped. Figure 3 is now Figure 5 in the revised manuscript. The new figure and its caption are reported in here in Figure R1 for the referee's perusal.

Figure R1: Downtime detection in real-time ($d_{RT}(t)$) for Kathmandu, Nepal. **(a)** Data are cropped at regular intervals to simulate real-time data collection (crossed markers along solid line). Square markers at $(t, d_{RT}(t))$ indicate the real time estimates for data cropped at $t = 30, 60, 90$ days. Dotted line indicates the ‘ideal’ downtime, $d(t)$, estimated using all data (see Section 3.3). **(b-d)** Black solid lines denote the cropped data used to estimate downtime at $t = 30, 60, 90$ days after the event (dashed magenta line) respectively. The cutoff t is shown with a black dashed line. The predicted downtimes $d_{RT}(t)$ are shown with coloured areas.

Point 8. *To apply the author’s methodology, how long does a business have to be in operation prior to the event? This seems to be an important since a newer business might not have enough data to apply this methodology accurately (given that the time series of each business is independent as discussed in comment 8.)*

Response: This is an interesting point and we thank the referee for prompting us to further investigate this aspect. In our analysis we consider all businesses that posted at least once, irrespective of the date of the event, hence we also include new businesses that started to post after the disaster. The rationale behind this choice is to use the same methodology to transform the entire time series, without differentiating between periods before and after the disaster. As the referee suggests, computing the downtime considering only businesses posting since at least one year before the date of the event may produce more robust results because businesses with a long posting history have more data and better statistics. A possible downside of this approach is that downtime estimates could be less accurate because of the fewer businesses considered, as explained in Section 3.2 about sensitivity analysis. Applying

our methodology just to businesses that posted one year before the events in the three regions, we get downtime estimates similar to those obtained looking at all businesses (Figure R2). In particular, the number of businesses remaining after the filtering is 2,781 in Kathmandu, 6,616 in San Juan and 380 in Juchitán de Zaragoza. The estimated lengths of downtime after the natural disasters are 48 days in Kathmandu, 91 days in San Juan, and 42 days in Juchitán de Zaragoza. The estimate for Kathmandu is the same as the estimate using all the businesses, while the estimated downtimes are shorter for San Juan and Juchitán de Zaragoza. As described in the sensitivity analysis section, a reduction of the downtime is expected when fewer businesses are considered. Note that in San Juan we detect an additional downtime of 17 days during the Christmas/New Year period, only 6 days after the end of the downtime due to Hurricane Maria. Combining these two downtimes we obtain an overall length of 114 days, which is very close to the 118 days estimated using all businesses.

We have included the results of this analysis (Figure R2) in the Section “Sensitivity analysis” in the Supplementary Materials, together with the following text:

In our analysis we consider all businesses that posted at least once, irrespective of the date of the event, hence we also include new businesses that started to post after the disaster. The rationale behind this choice is to use the same methodology to transform the entire time series, without differentiating between periods before and after the disaster. Computing the downtime considering only businesses posting since at least one year before the date of the event may produce more robust results because businesses with a long posting history have more data and better statistics. A possible downside of this approach is that downtime estimates could be less accurate because of the fewer businesses considered. Applying our methodology just to businesses that posted one year before the events in the three regions, we get downtime estimates similar to those obtained looking at all businesses. In particular, the number of businesses remaining after the filtering is 2,781 in Kathmandu, 6,616 in San Juan and 380 in Juchitán de Zaragoza. The estimated lengths of downtime after the natural disasters are 48 days in Kathmandu, 91 days in San Juan, and 42 days in Juchitán de Zaragoza. The estimate for Kathmandu is the same as the estimate using all the businesses, while the estimated downtimes are shorter for San Juan and Juchitán de Zaragoza. As shown in Figure S7, a reduction of the downtime is expected when fewer businesses are considered. Note that in San Juan we detect an additional downtime of 17 days during the Christmas/New Year period, only 6 days after the end of the downtime due to Hurricane Maria. Combining these two downtimes we obtain an overall length of 114 days, which is very close to the 118 days estimated using all businesses.

Figure R2: Downtime considering businesses just posting 1 year before the event date. Disaster event is shown in red, with other detected events shown in blue. The date of the event has been highlighted in magenta.

Point 9. *In data processing on page 8, the authors note that there are seasonality effects. How, exactly, is the seasonality dealt with? The authors' correction seems to be only for a linear trend.*

Response: We agree with the referee that the original sentence about the seasonality effects was misleading. Seasonality effects are still present in the transformed time series $r_U(t)$. For example, we can observe downtime during Dashain festivals in Kathmandu and during Christmas and New Year in Puerto Rico. The fact that downtime events are detected during these holiday periods provides further evidence of the ability of the proposed methodology to detect periods of reduced activity of businesses. Moreover, we would like to emphasize that our methodology is not limited to removing a linear trend, which is just part of Step 3 of our 4-step methodology. The main goals of the methodology are: (i) to remove the long-term nonlinear trend from the time series of aggregated posting activity, while retaining the dynamics at short and medium time scales (i.e. weeks, months); and (ii) to equally weight the contribution of each business in the aggregated time series, avoiding to over-represent the activity of businesses with higher posting rates. The outcome, $r_U(t)$, is a transformed time series without any long-term (over years) trend and without bias towards highly active businesses.

To clarify the aims and outcomes of the methodology we have removed from the main text the part related to the seasonality effects and added the following paragraph to Section 2.2.4:

The proposed methodology allows to: (i) remove the long-term nonlinear trend from the time series of aggregated posting activity, while retaining the dynamics at short and medium time scales (i.e. weeks, months); and (ii) equally weight the contribution of each business, avoiding to over-represent the activity of businesses with higher posting rates. The outcome is a transformed time series without any long-term trend and without bias towards highly active businesses.

Point 10. *In step 2 on page 9, the authors are using the Irwin-Hall distribution (and eventually the central limit theorem) which requires uncorrelated random variables. How are the authors justifying the variable independence? The correction discussed in step 3 does not adjust for correlation, only a linear trend. Moreover, this method hinges on the idea that the businesses' Facebook activity is highly correlated after an emergency.*

Response: The referee has correctly identified that the methodology relies on the activity of the businesses to be correlated after the emergency in order to observe a period when the aggregated posting activity is significantly lower than expected. In fact, whenever businesses have correlated behaviour the method is able to detect a period of anomalous activity, for instance during national holidays when most businesses are closed.

Under normal conditions we expect businesses to have independent posting activities. To be more precise, this is our definition of “normal conditions”. Indeed, when businesses' posting activities are similar to independent random variables, the assumptions behind the Irwin-Hall distribution and the central limit theorem hold and the transformed time series $r_U(t)$ will be fluctuating uniformly around 0.5. Instead, when activities are highly correlated, the independence assumption does not hold and large fluctuations will be present in $r_U(t)$, which are the signature of an anomalous behaviour that we would like to detect using the ‘elbow method’.

To clarify this point we added the following sentences in the Method section in subsection ‘Step 2: Shift and rescale.’:

Under normal conditions (i.e. non-emergency and non-holiday periods), we expect businesses to have independent posting activities. Specifically, this is our definition of ‘normal conditions’: when businesses' posting activities can be considered independent random variables.

and the following part in subsection ‘Downtime detection’:

In normal conditions the transformed time series $r_U(t)$ is expected to fluctuate uniformly within 0 and 1. In anomalous conditions, for example when all businesses are closed after a disaster, their posting activities are instead highly correlated. In these cases, large fluctuations will be present in $r_U(t)$, which are

the signature of an anomalous behaviour (i.e. a period of downtime) that we would like to detect using the ‘elbow method’ described in this section.

Referee #2

This manuscript provides analysis of Facebook posting volumes to infer the time of closure of businesses after natural disasters. The focus of the work is primarily on an algorithm to detect changes in time series of the number of posts made by businesses in a specified geographic area. While the topic is interesting, the results feel anecdotal and the manuscript does not give substantive insight into general conclusions from this work. A few specific concerns are noted below:

Response: We are pleased to see that the reviewer considers our work to be of interest. We have reorganised the overall structure of the manuscript in a more systematic way. Firstly, we present the methodology in its general form. Secondly, results are presented for the three case studies selected. The updated structure better highlights the general applicability of the proposed methodology in a wider context beyond the three case studies considered.

New validation data have been added, as suggested by the referee in point 4, and the new structure of the validation section provides substantive insights into general conclusions.

We have answered all the points raised by the referee in the following detailed responses.

Point 1. *It is unclear what exactly is to be concluded from the study. The first paragraph concludes with some speculation that the results might be useful to "target their interventions" and "distribute the available resources more effectively", but the metrics considered here don't link in any obvious way to those topics. Rather, the focus is primarily on the proposed algorithm and its application to three time series.*

Response:

The following conclusions can be drawn from the study:

- We demonstrate that the posting activity of small businesses on social media can be used to accurately estimate the recovery status of regions hit by natural disasters in real time.
- The methodology has general applicability: it works for different types of natural disasters (earthquakes and hurricanes) and for regions in different continents, in both developed and developing countries.
- The proposed framework offers several advantages with respect to other methods to estimate long-term post-disaster economic recovery:
 - it is cheaper and more scalable than traditional field surveys and interviews [1];

- it directly relates to human activity rather than to quantify physical damage, as common in many approaches based on satellite imagery [2, 3].
 - it is privacy-compliant because it is not based on sensitive personal data, in contrast with approaches using individual mobility and location data to measure the displacement of the affected population [4, 5, 6];
 - it is significantly easier to implement than methods based on text and sentiment analysis of social media content [7, 8, 9, 10, 11], which require the use of sophisticated natural language processing algorithms that should be tailored to the specific regions under investigation to prevent biased estimates, as highlighted in the paper mentioned by the referee [12].
- With respect to future applications, an automatic downtime detection system can be built using the proposed methodology and help with the identification of regions that are struggling to recover from natural disasters. The construction of such automatic downtime detection system is possible because of the method's unique features, which make the proposed framework highly scalable:
 - global coverage: The method has no intrinsic geographic coverage limitation; it is applicable to any region of the world struck by natural disaster in which a sufficient number of businesses (i.e. more than 1,000) are active on social media.
 - easy implementation: it generates real-time downtime estimates completely automatically, without the need to make any ad-hoc or manual adjustment to the algorithm.

The information provided by this automatic downtime detection system would be of interest to various stakeholders aiming to provide support to regions hit by disasters, including local and national governments, international financial institutions (e.g. the World Bank), and humanitarian organisations. One possible use of the system would be to identify areas whose recovery appears to be lagging behind, in order to deliver help and support where it is most needed.

- The proposed methodology for the detection of anomalous events in non-stationary time series is of general applicability to all cases where the main signal is composed by the aggregation of a large number of individual components, for example phone users' calling activities or internet users' visits to web pages.

To better clarify the conclusions that can be drawn from the study we included the points presented above in the Discussion.

Point 2. *It is unclear how the specific regions of study were determined here. Disaster impacts vary spatially, and presumably there are hard-hit regions and less impacted regions in each study area, but it is unclear what degree of impact was seen in each of these study*

areas, or how the closure time would be affected by changes in the study extent. Palen and Anderson (2016) note the importance of this decision—a reference that the authors should review in general.

Response:

We thank the reviewer for bringing Palen and Anderson’s article [12] to our attention. In their paper they make the following suggestion regarding the choice of the boundary of observation:

“To isolate activity by location or with respect to new and unusual behaviors, data sets must get bigger (by collecting contextual streams), before they can be sampled or filtered accordingly. This is because there are few natural constraints on social media data. There is no automatic mechanism for drawing one’s ‘unit of analysis’ and scope.”

We do agree that collecting as much data as possible, without applying any filter *a priori*, is the best way to avoid introducing biases in the analysis. We also agree that there is no automatic mechanisms for drawing the unit of analysis and scope. In the three regions analysed in our work, we drew bounding boxes around the main urban areas of interest in order to include the city centre and most part of the urban extent.

As explained in the Discussion, our method is highly scalable and in principle could be used to monitor entire regions. As suggested by Palen and Anderson, it would then be possible to “*Make ‘Big’ Data Bigger, Then Smaller*” and isolate specific regions to analyse the impact locally. As for the smaller region size that can be reliably analysed, we show in the Section Sensitivity analysis that our method gives consistent estimates of downtime when the number of businesses is around 1,000 or above.

Another important point made in [12] is that “*Social media data do not necessarily represent all of a population evenly, but they do represent a range of behaviors, ideas, and opinions that have a role to play alongside traditional disaster response.*” We are aware that not all businesses may have a social media account and the group that does have it may not be representative of the entire business population. When using the proposed methodology to draw conclusions on the degree of impact in different regions, particular attention should be taken to ensure to have a representative sample of the businesses in the region. Should that not be the case, the downtime for the type of businesses not represented in the data set should be estimated using other methods.

To clarify this point we added the following sentence in the Discussion:

Like in all studies based on social media data, it is important to remember that such data may not necessarily represent all of a population evenly [12]. In fact, some businesses may not have a social media account and the group that does have it may not be representative of the entire business population. When using the proposed methodology to draw conclusions on the degree of

impact in different regions, particular attention should be taken to ensure to have a representative sample of the businesses in the region. Should that not be the case, the downtime for the type of businesses not represented in the data set should be estimated using other methods.

Point 3. *It is unclear what it means for a region to have a single "downtime" number. The recovery process is a continuous one, and certainly the businesses are re-opening individually over time. Without a well-defined metric for what "downtime" is, the interpretation of the reported results, and the opportunities for validation of the results, are quite difficult.*

Response:

The length of downtime of a system is generally defined as the length of time during which it is not operating as expected, i.e. the level of a given indicator of the system's performance or activity is significantly reduced [13]. In the context of estimating the downtime of small businesses in a region, we define the indicator of *aggregated* activity for the region as the transformed time series of the number of posts made by all businesses, $r_U(t)$ (see Section 2.2.4). The transformed time series $r_U(t)$ is a good indicator of the aggregated activity of all businesses because it has no long-time (yearly) trend and it equally weights the activity of each business.

We then identify periods of significantly reduced aggregated activity (i.e. downtime) applying the 'elbow method' (defined in Section 2.3) to the time series $r_U(t)$. The 'elbow method' allows to determine a threshold on the level of activity, T^* , where activity fluctuations become significantly and unusually large. Hence, the length of downtime is estimated as the time taken for $r_U(t)$ to return to a normal level of activity, above the threshold T^* .

We gave a more specific definition of downtime in Section 2.3 Downtime detection:

The length of downtime of a system is generally defined as the length of time during which it is not operating as expected, i.e. the level of a given indicator of the system's performance or activity is significantly reduced [13]. In the context of estimating the downtime of small businesses in a region, we define the indicator of *aggregated* activity for the region as the transformed time series of the number of posts made by all businesses, $r_U(t)$ (see Section 2.2.4). The transformed time series $r_U(t)$ is a good indicator of the aggregated activity of all businesses because it has no long-time (yearly) trend and it equally weights the activity of each business. To identify periods of significantly reduced aggregated activity (i.e. downtime) we propose the 'elbow method', which allows to determine a threshold on the level of activity, T^* , where activity fluctuations become unusually large. The length of downtime is then estimated as the time taken for $r_U(t)$ to return to a normal level of activity, above the threshold T^* .

Point 4. *Related to the prior comment, the benchmarking of these results is weak. While it is admittedly difficult to get ground truth of closure times, it is concerning that each case study has a different method for determining an actual downtime—presumably different methods applied to a single disaster event would produce differing downtime estimates. The benchmarks are also based on extremely limited data (e.g., 16 survey responses), or very indirect measures (e.g., using tourist arrival numbers). It seems feasible to do something more direct and reproducible such as checking the posts for messages of "we are open again" or similar. The data sets are not huge, and sampling schemes or text analysis algorithms could help.*

Palen, L., and Anderson, K. M. (2016). "Crisis informatics—New data for extraordinary times." Science, 353(6296), 224–225.

Response: We agree that the benchmarking of downtime is difficult, and we recognise that the validation section in the original manuscript did not fully articulate the variety of sources used. In the revised manuscript, we find more validation sources of the actual downtime (i.e. official reports, scientific publications, text analysis, surveys), showing that they all agree on similar downtime estimates. We see the successful validation of our methodology against different data sources as a strength, rather than a weakness. A detailed list of all validation approaches used in each region is now provided in Table 1, which also appears in Section 3.1 in the manuscript.

We thank the reviewer for the suggestion to look directly at the contents of the messages, allowing for a uniform validation approach for all of the considered regions. To this extent we have sampled posts in each region, and searched for keywords indicating resumed business activity. The keywords that were chosen were 'open again', 'reopen', 'normal', 'regular', in English, Spanish and Nepalese. This method of validation has been added to the manuscript with the following text:

Text analysis of social media content has been previously used to assess the impact of natural hazards on different communities [7, 12]. We employ a text analysis method to obtain information about the reopening date of the businesses looking at the content of their messages on Facebook. We obtain the average reopening times reported by businesses using the following steps:

1. **Sample posts following the natural disaster.** We look at posts published up to five months from the date of the event (six months in San Juan). We sampled 40,946 posts in Kathmandu, 94,611 posts in San Juan and 4,536 posts in Juchitán de Zaragoza respectively.
2. **Select keywords.** We determine a set of keywords that indicate that a business has reopened. We select all messages containing the words: 'open again', 'reopen', 'normal' and 'regular', in English, Spanish and Nepalese.

Examples of posts retrieved include: *'we are back to work life is becoming normal'* and *'We have resumed to our regular operation.....!!!'*.

3. **Estimate the reopening date of each business.** For each business in our sample, we estimate its reopening date as the date of the first post that contains one of the selected keywords.
4. **Calculate average downtime.** The overall downtime is defined as the average of the business downtimes.

This method is applicable to all three regions considered. Our downtime estimates using Facebook posts text analysis are: 51 days for Kathmandu, 71 days for Puerto Rico and 55 days for Juchiatán de Zaragoza. The text analysis' results for Kathmandu and Juchiatán de Zaragoza are very similar to the lengths of downtime estimated with our method, while a shorter downtime is obtained for San Juan. Some limitations of the text analysis for estimating the downtime are discussed in the Supplementary Materials.

The text analysis' results for Kathmandu and Juchiatán de Zaragoza are very similar to the lengths of downtime estimated with our method, while a shorter downtime is obtained for San Juan. To validate the automatic text analysis approach, we manually read 19,928 posts from businesses in Kathmandu and estimated the reopening date of each business based on the context of all of its posts. We obtained a downtime of 50 days, which is compatible with the 51 days of the automatic text analysis algorithm.

On the other hand, the application of the text analysis approach emphasised some of the limitations of such kind of analyses, as discussed in [12], which we report in the Supplementary Materials:

Ambiguity about recovery status. We found cases where businesses state that they will reopen, and then never post again. It is unclear here whether they actually recovered.

Businesses do not post whether they have reopened. The majority of businesses do not explicitly state that they are open - where we have collected 40946 posts in Kathmandu, 94611 posts in San Juan and 4536 posts in Juchitán de Zaragoza respectively, there were only hundred of posts containing the keywords that were used to filter the messages.

Keywords are difficult to establish. Keywords are specific to each region. Local dialects and slang make the task of identifying relevant keywords difficult when validating this data using Facebook posts.

Repeated posts about recovery status. Additionally, businesses who do say they have reopened often repeatedly posts that they have reopened. To deal with this case only the first posts to mention a keyword for each businesses is used for analysis.

In summary, we applied the same validation method, i.e. posts text analysis, on all three regions obtaining estimates compatible with those of the proposed method.

We also showed that several validation methods based on different data sources previously used in the literature to estimate downtime recovery (e.g. [5, 14]) produce similar downtime estimates, which are compatible with those of the proposed method.

Region and Event	Source	Downtime Length
Kathmandu, Nepal Gorkha Earthquake	Estimated downtime	48 days
	Business surveys, from [3]	41 days
	Kathmandu Post Disaster Needs Assessment [15]*	37 days
	Mobile phone data, from [5]	56 days
	Facebook posts text analysis (n = 299)	51 days
San Juan, Puerto Rico Hurricane Maria	Estimated downtime	118 days
	Satellite imagery, from [14, 16]	134 days
	Puerto Rico Tourism Company ¹	97 days
	U.S. Energy Information Administration²	128 days
	Facebook posts text analysis (n = 755)	71 days
Juchitan de Zaragoza, Mexico Chiapas Earthquake	Estimated downtime	52 days
	Facebook surveys (n = 16)	63 days
	Facebook posts text analysis (n = 19)	55 days

Table 1: Downtimes estimated with the proposed framework (Estimated downtime) for the three natural disasters considered, along with the downtimes reported by the various validation sources described in Section 3.1. * “The earthquakes and series of continuing aftershocks led to the complete closure of schools and colleges for 37 days” (<https://www.nepalhousingreconstruction.org/sites/nuh/files/2017-03/PDNA%20Volume%20A%20Final.pdf>). ¹ <https://tourism.pr.gov>. ² <https://www.eia.gov/electricity/monthly/>.

The red rows denote new validation sources added in the revised version of the manuscript.

References

- [1] R. Campanella, “Street survey of business reopenings in post-katrina new orleans,” 2007.
- [2] E. Booth, K. Saito, R. Spence, G. Madabhushi, and R. T. Eguchi, “Validating assessments of seismic damage made from remote sensing,” *Earthquake Spectra*, vol. 27, p. S157–S177, Oct 2011.

- [3] F. De Luca, D. Aldamen, J. Kurton, M. Wray, R. Mohan Pokhrel, and M. Werner, “Traffic data as proxy of business downtime after natural disasters: The case of kathmandu,” in *11th National Conference on Earthquake Engineering*, 6 2018.
- [4] X. Lu, L. Bengtsson, and P. Holme, “Predictability of population displacement after the 2010 haiti earthquake,” *Proceedings of the National Academy of Sciences*, vol. 109, p. 11576–11581, Jun 2012.
- [5] R. Wilson, E. zu Erbach-Schoenberg, M. Albert, D. Power, S. Tudge, M. Gonzalez, S. Guthrie, H. Chamberlain, C. Brooks, C. Hughes, and et al., “Rapid and near real-time assessments of population displacement using mobile phone data following disasters: The 2015 nepal earthquake,” *PLoS Currents*, 2016.
- [6] P. Maas, S. Iyer, A. Gros, W. Park, L. McGorman, C. Nayak, and P. A. Dow, “Facebook disaster maps: Aggregate insights for crisis response & recovery,” in *Proceedings of the 25th ACM SIGKDD International Conference on Knowledge Discovery & Data Mining*, pp. 3173–3173, ACM, 2019.
- [7] D. Murthy and A. J. Gross, “Social media processes in disasters: Implications of emergent technology use,” *Social Science Research*, vol. 63, p. 356–370, Mar 2017.
- [8] L. Zou, N. S. N. Lam, H. Cai, and Y. Qiang, “Mining twitter data for improved understanding of disaster resilience,” *Annals of the American Association of Geographers*, vol. 108, p. 1422–1441, Mar 2018.
- [9] P. S. Earle, D. C. Bowden, and M. Guy, “Twitter earthquake detection: earthquake monitoring in a social world,” *Annals of Geophysics*, vol. 54, no. 6, 2012.
- [10] S. Vieweg, A. L. Hughes, K. Starbird, and L. Palen, “Microblogging during two natural hazards events,” *Proceedings of the 28th international conference on Human factors in computing systems - CHI '10*, 2010.
- [11] J. Sutton, E. S. Spiro, B. Johnson, S. Fitzhugh, B. Gibson, and C. T. Butts, “Warning tweets: Serial transmission of messages during the warning phase of a disaster event,” *Information, Communication & Society*, vol. 17, no. 6, pp. 765–787, 2014.
- [12] L. Palen and K. M. Anderson, “Crisis informatics—new data for extraordinary times,” *Science*, vol. 353, no. 6296, pp. 224–225, 2016.
- [13] S. E. Chang, “Urban disaster recovery: a measurement framework and its application to the 1995 kobe earthquake,” *Disasters*, vol. 34, p. 303–327, Mar 2010.
- [14] J. Shermeyer, “Assessment of electrical and infrastructure recovery in puerto rico following hurricane maria using a multisource time series of satellite imagery,” in *Earth Resources and Environmental Remote Sensing/GIS Applications IX* (U. Michel and K. Schulz, eds.), SPIE, Oct 2018.

- [15] N. P. Commission *et al.*, “Post disaster needs assessment,” *Vol. A and B. Kathmandu: Government of Nepal*, 2015.
- [16] J. Shermeyer, “Comet time series visualizer: Cometts,” *Journal of Open Source Software*, vol. 4, p. 1047, Oct 2019.

REVIEWERS' COMMENTS:

Reviewer #1 (Remarks to the Author):

The authors have satisfactorily addressed my comments.

Reviewer #2 (Remarks to the Author):

The revised manuscript is greatly improved, with the restructuring adding more logic to the manuscript and the additional benchmarking providing some evidence of robustness in the algorithm.

Authors' response:

“Social media usage reveals small businesses recovery
after natural hazard events”

Reviewer 1

The authors have satisfactorily addressed my comments.

Reviewer 2

The revised manuscript is greatly improved, with the restructuring adding more logic to the manuscript and the additional benchmarking providing some evidence of robustness in the algorithm.

Response

We are happy that both Reviewers find that the new version of the manuscript is greatly improved and we satisfactorily address all comments.